

# RNAi-based knockdown of candidate gut receptor genes altered the susceptibility of *Spodoptera frugiperda* and *S. litura* larvae to a chimeric toxin Cry1AcF

Tushar K. Dutta[1], Kodhandaraman Santhoshkumar[1], Arudhimath Veeresh[1], Chandramani Waghmare[1], Chetna Mathur[1] and Rohini Sreevathsa[2]

[1] Division of Nematology, ICAR-Indian Agricultural Research Institute, New Delhi, Delhi, India
[2] ICAR-National Institute for Plant Biotechnology, New Delhi, Delhi, India

## ABSTRACT

**Background:** A multitude of Cry toxins (secreted by *Bacillus thuringiensis* or Bt) has been deployed globally either *via* transgenic mean or bio-pesticidal formulations in order to manage insect pests. However, Bt resistance development in insects is emerging as a major concern. To avoid this problem, multiple gene pyramiding or protein-engineered chimeric toxin-based strategy has been analyzed.

**Methods:** In the present study, one such chimeric toxin Cry1AcF (contain the swapped domains of Cry1Ac and Cry1F) was used to investigate its *in vivo* pathogenesis process in lepidopteran pests *Spodoptera frugiperda* and *S. litura*. A number of biochemical and molecular analysis were performed.

**Results:** Oral ingestion of Cry1AcF caused greater toxicity in *S. frugiperda* than *S. litura* with larvae displaying increased hemolymph melanization. Histopathology of the midgut transverse sections exhibited Cry1AcF-induced extensive gut damage in both the test insects followed by cytotoxicity in terms of reduced hemocyte numbers and viability. Elevated hemolymph phenoloxidase activity indicated the immune-stimulatory nature of Cry1AcF. In order to analyze the role of gut receptor proteins in Cry1AcF intoxication in test insects, we performed RNAi-mediated silencing using bacterially-expressed dsRNAs of individual receptor-encoding genes including CAD, ABCC2, ALP1 and APN. Target-specific induced downregulation of receptor mRNAs differentially altered the insect susceptibility to Cry1AcF toxin in our study. The susceptibility of ALP1 and APN dsRNA pre-treated *S. frugiperda* was considerably decreased when treated with Cry1AcF in $LD_{50}$ and $LD_{90}$ doses, whereas susceptibility of CAD and ABCC2 dsRNA pre-treated *S. litura* was significantly reduced when ingested with Cry1AcF in different doses. CAD/ABCC2-silenced *S. frugiperda* and ALP1/APN-silenced *S. litura* were vulnerable to Cry1AcF alike of control larvae. In conclusion, our results indicate ALP1/APN and CAD/ABCC2 as the functional receptor for Cry1AcF toxicity in *S. frugiperda* and *S. litura*, respectively.

Corresponding author
Tushar K. Dutta,
tushar.dutta@icar.gov.in

## INTRODUCTION

According to a United Nations report, the current global human population (7.9 billion) may grow to 8.5 and 9.7 billion in 2030 and 2050, respectively (https://www.un.org/en/observances/world-population-day). This ever-growing population has put an asymmetric demand on food supply especially when global agriculture is facing shrinkage of cultivable land, water crisis and arrival of new pests and pathogens due to climate change (*IPPC Secretariat, 2021*). Although protected agriculture and vertical farming are rapidly being adopted, insect pests are emerging as a major problem due to the conducive environment maintained inside the protected structures (*Phani, Khan & Dutta, 2021*). Indian agriculture suffers heavily due to a multitude of insect pest occurrence including noctuid lepidopteran pests such as *Spodoptera litura* and *S. frugiperda* (*Dhaliwal, Jindal & Mohindru, 2015*; *EFSA Panel on Plant Health, 2019*; *Suby et al., 2020*). The polyphagous *S. litura* can infect over 100 crops and are widely distributed across tropical and subtropical Asia including India, China and Japan (*Cheng et al., 2017*). In Africa, America and Asia (including Indian subcontinent), the fall armyworm (*S. frugiperda*) has become an invasive species (mostly in cotton and cultivated grasses such as maize, sorghum, rice, sugarcane *etc.*) because of the global warming effects (*CABI Compendium, 2019*; *IPPC Secretariat, 2021*).

Synthetic pesticides are continually being withdrawn from market or placed under restricted use because of the insecticide resistance emergence in insects, lesser degradability of pesticides in soil, avian and mammalian toxicity of insecticide residues, *etc*. As a result, very few active insecticides are at disposal for the growers (http://ppqs.gov.in/). Use of biological control agents such as *Bacillus thuringiensis* (Bt) for pest control either *via* spraying or transgenic mean has been met with great success (*Koch et al., 2015*; *Liu et al., 2021*). Bt secretes an insecticidal crystalline protein popularly known as delta endotoxin or Cry toxin which kills the insect (*Bravo, Gill & Soberon, 2007*). Worldwide, the acreage of Bt transgenic crops has increased tremendously since their introduction during 1996 (*ISAAA, 2019*). Currently, a number of Bt crops are commercially grown including Bt maize, Bt cotton, Bt soybean, Bt rice and Bt eggplant (*Chakrabarty et al., 2022*). Cry intoxication mechanism may involve pore formation or cell signaling model. In either cases, the activated toxin has to bind with different receptors such as cadherin (CAD), aminopeptidase N (APN), alkaline phosphatase (ALP) and ATP-binding cassette (ABC) transporters located in the epithelial cell membrane of the insect midgut. Binding of Cry protein with these receptors lead to oligomerization of Cry monomers that ultimately lead to cytotoxicity (*Zhang et al., 2006*; *Adang, Crickmore & Jurat-Fuentes, 2014*; *Jurat-Fuentes, Heckel & Ferré, 2021*). Domain I of Cry toxin aids in tethering to the cell membrane *via* its hydrophobic helical hairpin, Domain II helps in detecting putative receptors *via* its highly variable loops and Domain III is associated with receptor binding and catalytic activity (*Chakrabarty et al., 2022*).

To date, a repertoire of Cry toxins has been characterized globally for their future deployment *via* transgenic mean. The sustained use of this technology was demonstrated when a number of those toxin genes were expressed in Bt crops (*Carrière et al., 2019*;

*Dutta, Phani & Mandal, 2022*). Nevertheless, reports of resistance development in insects towards different Cry protein encoding genes owing to wider deployment of narrow-spectrum Bt crops in India, China, Brazil, Argentina and United States have become an alarming concern (*Tabashnik, Brévault & Carrière, 2013*; *Tabashnik & Carrière, 2017*; *Carrière et al., 2019*). In order to minimize the Bt resistance development in insects, multiple gene pyramiding or protein-engineered chimeric toxin-based strategy has been advocated (*Honée, Vriezen & Visser, 1990*; *Ho et al., 2006*; *Tajne et al., 2014*; *Zghal et al., 2017*; *Zafar et al., 2020*; *Chen et al., 2021*). It is assumed that chimeric toxins would have broad-spectrum efficacy that would aid in averting Bt resistance development in insects (*Rathinam et al., 2019*). One such chimeric toxin was generated by fusing Domain I and II of Cry1Ac with Domain III of Cry1F and named as Cry1AcF, which is an activated toxin (patent no 237912; *Rathinam et al., 2017*). This chimeric toxin when expressed *via* transgenic mean in different pulse crops it showed resistance against the lepidopteran pest *Helicoverpa armigera* in confined trials (*Keshamma et al., 2012*; *Ramu et al., 2012*; *Muralimohan et al., 2020*; *Ramkumar et al., 2020*). However, information about the role of this toxin in insect *in vivo* pathogenesis was lacking. In our recent studies, we showed that Cry1AcF can negatively alter the midgut architecture of the model insect *Galleria mellonella* and the toxin requires to bind with different gut receptors to become insecticidal (*Dutta et al., 2022a*, *2022b*; *Dutta, Phani & Mandal, 2022*). In the present study, we analyzed the *in vivo* pathogenesis of Cry1AcF in economically important insects such as *S. frugiperda* and *S. litura*, and the putative role of gut receptors in Cry1AcF intoxication process was established.

## MATERIALS AND METHODS

### Insect rearing

*S. frugiperda* and *S. litura* egg clusters were obtained from infested plots (IARI, New Delhi) of maize and okra followed by initiation of nucleus colony culture in our laboratory. Rearing of neonates was performed in sterilized castor leaves in a growth chamber; sixth-instar larvae were pupated in sterilized sand followed by disinfection of pupae *via* rinsing in 0.02% formaldehyde. Adults were fed with artificial diets comprising honey, multivitamin supplements, ascorbic acid and methyl-p-hydroxybenzoate (*Shankhu et al., 2020*; *Dutta et al., 2021*). A continuous culture was maintained on castor leaves for recurrent insect life cycle progression. Surface-sterilized (with 70% ethanol) fourth-instar larvae (0.4–0.5 g body mass) were used for subsequent experiments.

### Cry1AcF toxicity assessment in test insects

Larvae were randomly assigned to each treatment. Larvae were starved for 6 h followed by oral ingestion with 10 µL of phosphate-buffered saline (PBS, pH 7.4) containing Cry1AcF toxin in different doses using a sterilized 26-gauge hypodermic needle (Hamilton syringe; Sigma-Aldrich St. Louis, MO, USA). Negative control consisted of PBS only. A known toxin TcaB (*Mathur et al., 2019*; *Dutta et al., 2021*) was used as the positive control. Larvae were incubated in sterile six-well polystyrene plates containing artificial diet at 28 °C, 60% relative humidity. Insect mortality data was recorded at 24 h after oral ingestion.

The whole experiment was repeated at least thrice (*n* = 50 per treatment). The details of Cry1AcF protein production and purification are described in our previous studies (*Rathinam et al., 2017*, *2019*).

## Histopathology study

At 24 h after oral ingestion, toxin-treated and control larvae were inoculated with 100 μL of 10% formalin *via* intra-hemocoel injection followed by dipping the larvae in formalin (*Shankhu et al., 2020*). After 24 h, midgut tissue was dissected out using a sterilized blade under the microscope, and suspended in ice-cold fixative (4% paraformaldehyde + 2.5% glutaraldehyde in 0.1 M PBS, pH 7.2) followed by transfer to fresh fixative and overnight storing at 4 °C. Next, samples dehydrated through a graded ethanol series. Dehydrated samples were immersed in xylene for clearing of tissues. Embedded (in paraffin wax) samples were processed into 6 μm cross-sections using Leica RM2165 microtome (*Dutta et al., 2021*; *Santhoshkumar et al., 2021*). Sections fixed in glass slides *via* egg albumin solution and dried overnight. Slides washed using graded ethanol series and sections dewaxed *via* xylene treatment. Sections were periodically stained with Cole's haematoxylin and eosin (Sigma-Aldrich, St. Louis, MO, USA). Samples were mounted in Dibutylphthalate Polysterene Xylene (DPX; Sigma-Aldrich St. Louis, MO, USA). More detail about the procedure can be found in *Dutta et al. (2021)* and *Santhoshkumar et al. (2021)*. Photomicrographs were obtained in an Zeiss Axiocam MRm microscope.

## Cry1AcF cytotoxicity analysis in test insects

Total hemocyte counts (THC) were obtained from toxin-treated and control larvae at 24 h after oral ingestion. Sample of fresh hemolymph (10 μL) was extracted (by piercing through the integument above head using a sterilized needle) from each larvae and diluted three-fold with ice-cold anticoagulant containing phenylthiourea (4 mg mL$^{-1}$ in PBS) to suppress the melanization (*Mathur et al., 2019*). To examine hemocyte viability, 0.4% Trypan blue (w/v in PBS) was mixed to the hemolymph, which was not treated with the anticoagulant (*Mathur et al., 2019*). On both occasions, hemolymph sample was applied into a hemocytometer (Neubauer) for cell enumeration in Zeiss Axiocam MRm microscope. Experiment was repeated thrice (*n* = 10 per treatment).

## Cry1AcF immune-stimulatory activity measurement in test insects

Phenoloxidase (PO) enzyme activity was measured in the hemolymph of toxin-treated and control larvae at 12 and 24 h after inoculation. A total of 10 μL hemolymph was extracted from each larvae as described above and 100 μL PBS was added to it by centrifugation at 1,006 g for 2 min at 4 °C. Cell-free supernatant was used for downstream analyses. A total of 100 μL of hemolymph sample (cell-free supernatant) was added to 100 μL of 20 mM L-3,4-dihydroxyphenylalanine or L-DOPA (4 mg mL$^{-1}$ in PBS) in a 96-well ELISA plate and incubated at 28 °C in dark for 30 min (*Mathur et al., 2019*; *Dutta et al., 2021*). Absorbance of the sample was examined at 490 nm in a microplate reader (BioTek, Winooski, VT, USA). PO activity was expressed as the increase in absorbance at 490 nm per minute per mg of protein. As control, PBS and L-DOPA were mixed equally and its

absorbance value was subtracted from each sample. Experiment was repeated thrice ($n$ = 10 per treatment).

## RNA extraction from *S. frugiperda* and *S. litura*

To analyze the transcription dynamics of different Cry receptor-encoding genes in different developmental stages and tissues of *S. frugiperda* and *S. litura*, RNA was extracted from different samples using TRIzol reagent (Invitrogen, Waltham, MA, USA) by following the manufacturer's protocols. In order to minimize genomic DNA contamination, extracted RNA was digested with DNase I enzyme (TakaRa, Kusatsu, Japan). Purity of the RNA was assessed in a Nanodrop ND-1000 spectrophotometer (Thermo Fisher Scientific, Waltham, MA, USA), and RNA integrity was evaluated by electrophoresing samples on 1% (w/v) agarose gel. RNA (~1 µg) was reverse transcribed to cDNA by using a first-strand cDNA synthesis kit (Superscript VILO; Invitrogen, Waltham, MA, USA).

## RT-qPCR analysis

RT-qPCR-based transcription profile analysis of Cry receptor-encoding genes was performed in a CFX96 thermal cycler (BioRad, Hercules, CA, USA). qPCR reaction volume (10 µL) consisted of 1.5 ng cDNA, forward and reverse primers (750 nM each), and 5 µL SYBR Green PCR master-mix (BioRad, Hercules, CA, USA). qPCR cycling conditions included a hot start phase of 95 °C for 30 s, followed by 40 cycles of 95 °C for 10 s and 60 °C for 30 s. Additionally, a melt curve program (95 °C for 15 s, 60 °C for 15 s, followed by a slow ramp from 60 °C to 95 °C) was included to ascertain the amplification specificity. Quantification cycle (Cq) values were obtained from CFX Maestro software (BioRad, Hercules, CA, USA). Housekeeping genes of *S. frugiperda* (ribosomal protein S3 or *rps3*) and *S. litura* (*β-actin* and *GAPDH*) were included as the internal reference for normalizing the expression value of target genes. qPCR fold change data was calculated using $2^{-\Delta\Delta Cq}$ method. qPCR was run with five biological and three technical replicates for each samples. Primer designing was carried out in OligoAnalyzer tool (https://eu.idtdna.com/). PCR reaction efficiency of different primers was determined by constructing a standard curve (where Cq values were plotted against cDNA concentrations) across the five-fold dilutions of cDNA samples followed by calculating the slope using a linear regression equation: $E = (10^{(-1/\text{slope})} - 1) \times 100$. Primer details and PCR efficiency data are given in Tables S1 and S2.

## DsRNA synthesis

Targeted region for dsRNA synthesis from the coding sequences of different receptors was designed using different *in silico* tools such as siDirect (http://sidirect2.rnai.jp/; predicts siRNA biogenesis probability across the query sequence), Dharmacon (http://horizondiscovery.com/) and dsCheck (http://dscheck.rnai.jp/; helps in avoiding off-target sites by comparing sequence homology of predicted siRNAs with that from non-target organism). DsRNA sequences were PCR-amplified (according to the standard protocol) from the cDNAs of *S. frugiperda* and *S. litura* (extracted from the midgut tissues
of fourth-instar larvae) using gene-specific primers that harbored *Sac*I and *Hind*III restriction endonuclease sites (Tables S1 and S2). PCR products were cloned into *Sac*I and *Hind*III-digested L4440 plasmid (Addgene) that contained two T7 polymerase promoter sequences (aids in driving DNA to RNA transcription) in inverted orientation flanking the multiple cloning site. DsRNA molecules were bacterially expressed in *Escherichia coli* HT115 (DE3) cells, which were transformed with recombinant L4440 plasmid. HT115 cells were cultured in Luria Bertani medium (Sigma, St. Louis, MO, USA) containing ampicillin (50 μg mL$^{-1}$) and tetracycline (12.5 μg mL$^{-1}$) at 37 °C for 12 h with constant agitation at 200 rpm. In order to induce T7 polymerase synthesis, 0.4 mM IPTG was mixed into the culture and incubated for an additional 4 h at 37 °C. Transcribed dsRNAs were isolated from bacterial aliquots (dsRNAs purified from *E. coli* cells by phenol/chloroform/isoamyl alcohol extraction *via* shaking at 65 °C for 30 min followed by centrifugation at 12,000 × g for 15 min, upper phase containing dsRNA transferred to a new tube, precipitated with isopropanol, washed with ethanol, followed by re-suspending the nucleic acid pellet in 0.05 M PBS) and electrophoresed on a 1% (w/v) agarose gel (Ahn et al., 2019). DsRNA of a *gfp* gene (Genbank ID: HF675000) ligated into L4440 plasmid was used as the non-native control (in order to check whether dsRNA itself has any negative effect on insects).

## RNAi bioassay

Bacterially-expressed dsRNAs (10 μL suspension amounting to ~10 μg of dsRNA) were orally administered into the 6 h starved fourth-instar larvae of *S. frugiperda* or *S. litura* using a sterilized needle as described above. GFP dsRNA and 0.05 M PBS served as the non-native and negative control, respectively. Inoculated larvae were incubated in sterile 6-well plates as described above. Post 24 h of dsRNA treatment, larvae were orally ingested with LD$_{50}$ or LD$_{90}$ doses of Cry1AcF toxin and incubated in six-well plates as depicted above. After 24 h insect mortality data was recorded. The whole experiment was repeated at least thrice ($n$ = 50 per treatment).

RNAi-induced downregulation (fold change in expression of a target gene in dsRNA-treated larvae was subtracted from that in control larvae) of target receptor-encoding genes was validated by RT-qPCR assay. RNA was isolated from the midgut tissues of ten random larvae belonging to each dsRNA treatment groups. RNA was reverse-transcribed to cDNA as depicted previously. qPCR run conditions were followed as explained above.

## Statistical analysis

LD$_{50}$ and LD$_{90}$ doses of Cry1AcF was determined by using probit analysis in SPSS v. 21 software (IBM Corp., Armonk, NY, USA). Data of different bioassay (THC and PO assay data were checked for normality using Shapiro–Wilk test) and qPCR experiments was subjected to one-way ANOVA followed by Tukey's honest significant difference (HSD) test (for multiple comparison among different treatments) in SAS v. 14.1 software (SAS, Inc. Cary, NC, USA).
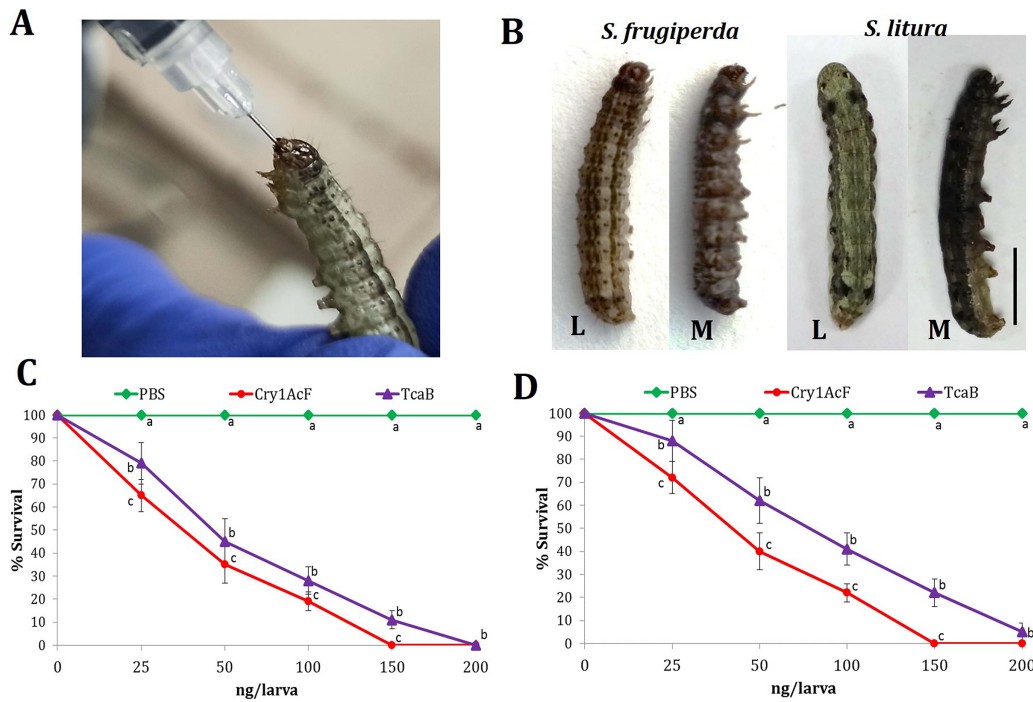

**Figure 1  Insecticidal activity of Cry1AcF toxin in *S. frugiperda* and *S. litura*.** (A) Oral ingestion of the toxin using hypodermic needle in starved *S. frugiperda* fourth-instar larvae. (B) At 24 h after inoculation, toxin (150 ng)-treated moribund (M) larvae attained dead-like posture with extended prolegs and hemolymph melanization compared to normal phenotypes of PBS-ingested control (L) larvae (scale bar—0.5 cm). Dose-response curves depict the percent survival of *S. frugiperda* (C) and *S. litura* (D) at 24 h after toxin ingestion. TcaB and PBS were used as the positive and negative control, respectively. X- and Y-axis represent toxin dose and percent larval survival, respectively. Treatments (mean ± SE, $n = 50$) with different letters are significantly different at $P < 0.01$, Tukey's HSD test.

## RESULTS

### Biological activity of Cry1AcF against *S. frugiperda* and *S. litura*

Oral ingestion of Cry1AcF (Fig. 1A) caused toxicity in the fourth-instar larvae of *S. frugiperda* and *S. litura*. Phenotypic changes such as dead-like posture with extended prolegs and hemolymph melanization was evident in Cry1AcF-treated larvae compared to normal phenotypes in PBS-ingested control larvae at 24 h after inoculation (Fig. 1B). Cry1AcF conferred a gradual and significant ($P < 0.01$) decrease in mean percent survival of *S. frugiperda* ($F (4, 45) = 56.82$, $P = 0.0021$) and *S. litura* ($F (4, 45) = 47.38$, $P = 0.0032$) with increasing Cry1AcF doses, *i.e.*, 25–200 ng/larva (Figs. 1C, 1D). While compared with a gut-active reference toxin TcaB (characterized from an insect parasitic bacterium *Photorhabdus akhurstii* in our laboratory; *Mathur et al., 2019*; *Santhoshkumar et al., 2021*), Cry1AcF caused significantly ($P < 0.01$) greater larval mortality across the dose range in both the test insects (Figs. 1C, 1D). Consequently, Cry1AcF exhibited lower $LD_{50}$ and $LD_{90}$ values than TcaB in both the test insects (Table 1). The $LD_{50}$ and $LD_{90}$ values for Cry1AcF in *S. frugiperda* was derived as 29.78 and 99.77 ng/larva, respectively. The $LD_{50}$ and $LD_{90}$ values for Cry1AcF in *S. litura* was calculated as 38.76 and 117.32 ng/larva, respectively

**Table 1 The $LD_{50}$ and $LD_{90}$ values of Cry1AcF toxin in *S. frugiperda* and *S. litura* fourth-instar larvae at 24 h after oral ingestion.** A known bacterial toxin TcaB was used as the reference. Values were determined by Probit analysis. Numbers in parentheses represent 95% confidence interval. $R^2$ value indicates the closeness of data to the fitted regression line.

| | | $LD_{50}$ | $LD_{90}$ | $R^2$ value |
|---|---|---|---|---|
| *S. frugiperda* | Cry1AcF (ng/larva) | 29.78 [15.55–39.54] | 99.77 [78.89–121.78] | 0.922 |
| | TcaB (ng/larva) | 53.89 [41.33–69.72] | 148.49 [127.85–170.43] | 0.918 |
| | Cry1AcF (ng/g)* | 53.60 | 179.59 | – |
| | TcaB (ng/g)* | 97.00 | 267.28 | – |
| *S. litura* | Cry1AcF (ng/larva) | 38.76 [22.88–51.67] | 117.32 [93.66–133.65] | 0.903 |
| | TcaB (ng/larva) | 69.86 [55.98–88.39] | 172.19 [146.75–191.48] | 0.925 |
| | Cry1AcF (ng/g)* | 69.77 | 211.18 | – |
| | TcaB (ng/g)* | 125.75 | 309.94 | – |

**Note:**
* The average larval mass during inoculation was ~0.45 g.

(Table 1). When larval body mass was included in calculation, Cry1AcF caused greater toxicity in *S. frugiperda* in terms of lower $LD_{50}$ (53.60 ng/g) and $LD_{90}$ (179.59 ng/g) values compared to *S. litura* ($LD_{50}$: 69.77 ng/g; $LD_{90}$: 211.18 ng/g).

## Cry1AcF causes gut leakiness in *S. frugiperda* and *S. litura*

Presumably, orally delivered Cry1AcF catalytically disrupted the midgut architecture of test insects leading to leaky gut and consequent hemolymph melanization. In order to validate this hypothesis, comparative histopathology experiment of PBS- and Cry1AcF-ingested (in $LD_{50}$ doses) *S. frugiperda* and *S. litura* midgut transverse sections was carried out at 24 h after inoculation. Midgut tissues of PBS-ingested larvae exhibited normal morphology in terms of regular arrangement of epithelial cells (with intact nucleus) which were resting on the basement membrane. Gut tissue damage was negligible because the protective barrier provided by visceral muscle layer (interface between gut epithelium and hemocoel or body cavity) was intact in both the test insects (Figs. 2A, 2C). By contrast, in Cry1AcF-ingested larvae, epithelial cell lining was displaced, cells were disintegrated and sloughed off into the gut lumen. Additionally, a discontinuous lining of visceral muscle layer was evident suggesting the structural damage to gut-body cavity barrier in the test insects (Figs. 2B, 2D).

## Cry1AcF reduces hemocyte numbers and viability in *S. frugiperda* and *S. litura*

Interestingly, the cytotoxic effect of Cry1AcF was observed in both the test insects at 24 h after oral ingestion. Total circulatory hemocyte counts (THC) of *S. frugiperda* was significantly ($F (2, 27) = 24.44$, $P = 0.0072$) reduced from $5.99 \times 10^6$ mL$^{-1}$ in PBS to $3.48 \times 10^6$ mL$^{-1}$ and $2.27 \times 10^6$ mL$^{-1}$ in Cry1AcF-treated insects at $LD_{50}$ (30 ng/larva) and $LD_{90}$ (100 ng/larva) doses, respectively. Similarly, THC of *S. litura* was significantly ($F (2, 27) = 29.37$, $P = 0.0058$) declined from $9.88 \times 10^6$ mL$^{-1}$ in PBS to $6.89 \times 10^6$ mL$^{-1}$ and 4.36
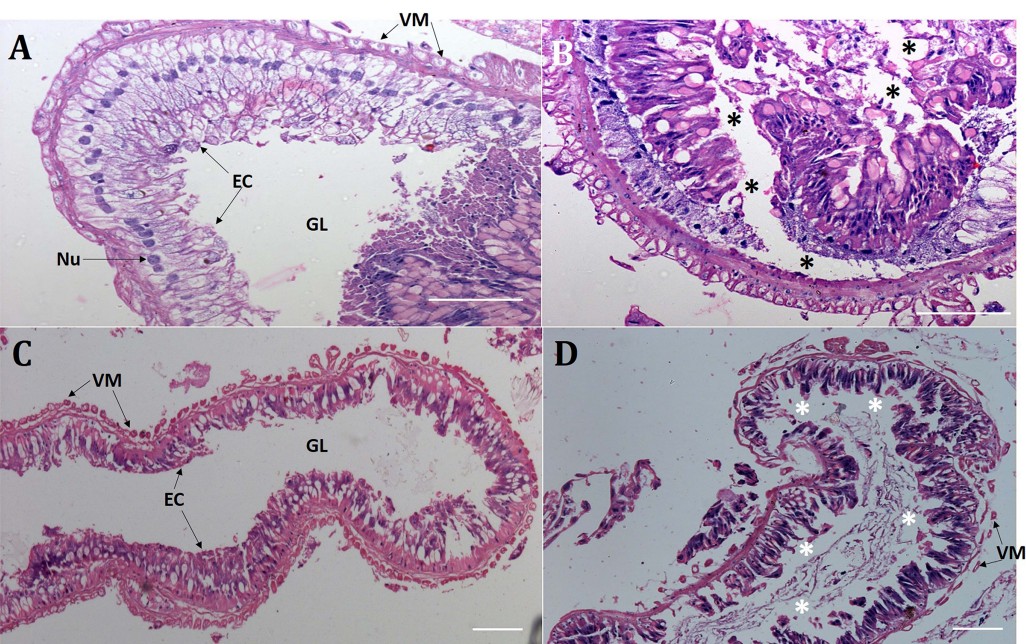

**Figure 2 Cry1AcF negatively altered the gut homeostasis of *S. frugiperda* and *S. litura* fourth-instar larvae.** Histopathology of the midgut transverse sections of *S. frugiperda* treated with PBS (A), Cry1AcF (B) and *S. litura* treated with PBS (C), Cry1AcF (D) at 24 h after inoculation. Cry1AcF was administered with $LD_{50}$ doses. Asterisks (*) indicate disintegration of epithelial cells (EC) followed by sloughing off into the gut lumen (GL). VM, visceral muscle; Nu, Nucleus. Scale bar = 50 µm.

$\times 10^6$ mL$^{-1}$ in Cry1AcF-treated insects at $LD_{50}$ (40 ng/larva) and $LD_{90}$ (120 ng/larva) doses, respectively (Fig. 3A).

In addition, viability of hemocyte cells was examined by Trypan blue staining assay. In this assay, viable cells avert the stain penetration while dead cells dye dark blue because of stain penetration *via* disintegrated cell membrane. Approximately 100 cells were sampled from each treatment ($n = 10$), and live cells were counted and expressed as percent viability. Cell viability of *S. frugiperda* was significantly ($F (2, 27) = 16.77$, $P = 0.0091$) reduced from 100% in PBS to 65% and 43% in Cry1AcF-treated insects at $LD_{50}$ and $LD_{90}$ doses, respectively. Likewise, cell viability of *S. litura* was significantly ($F (2, 27) = 14.88$, $P = 0.0062$) decreased from 100% in PBS to 69% and 49% in Cry1AcF-treated insects at $LD_{50}$ and $LD_{90}$ doses, respectively (Fig. 3B). Comparative photomicrographs (generated *via* Trypan blue exclusion assay) represent the live and dead hemocyte cells of *S. frugiperda* treated with PBS and Cry1AcF, respectively (Figs. 3C, 3D).

## Cry1AcF induces immune-stimulatory activity in *S. frugiperda* and *S. litura*

The disruption of hemocyte or immunocyte cell morphology due to Cry1AcF-induced toxicity is indicative of Cry1AcF immune-stimulatory activity in test insects. Additionally, larval melanization due to Cry1AcF catalytic activity exemplifies the elevated PO enzyme activity in the hemolymph resulting from the conversion of hemocyte-bound proPO to

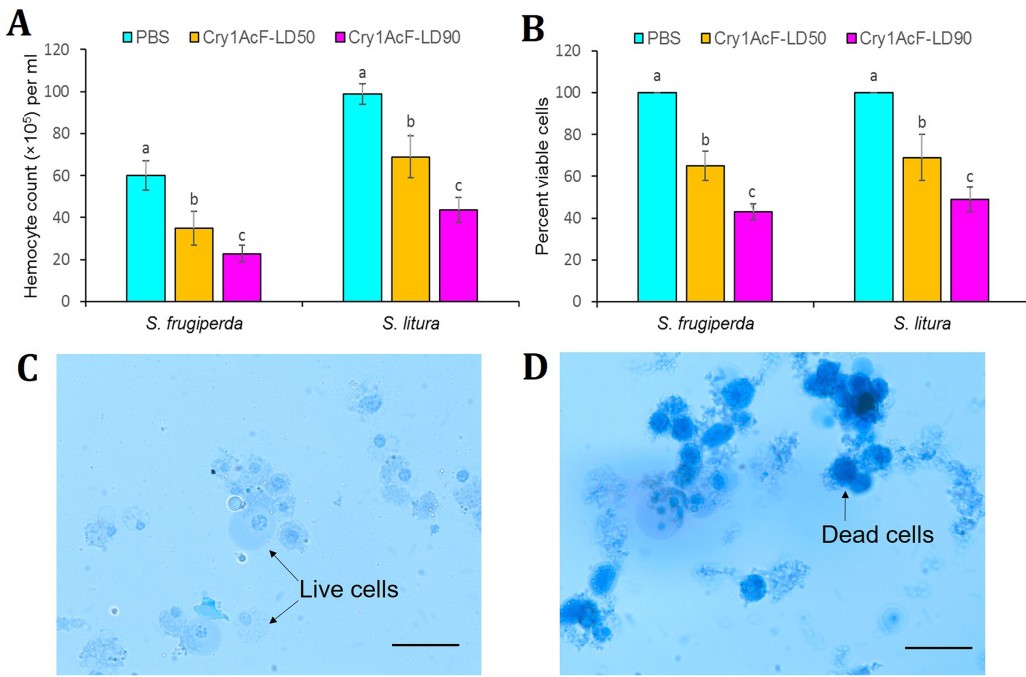

**Figure 3 Effect of Cry1AcF toxin on the total circulatory hemocytes of *S. frugiperda* and *S. litura* fourth-instar larvae at LD$_{50}$ and LD$_{90}$ doses.** (A) Y-axis indicates hemocyte counts (×10$^5$) per ml of hemocoel extracted at 24 h after toxin ingestion. (B) Y-axis indicates percent viable cells (determined by Trypan blue staining assay) at 24 h after toxin ingestion. PBS was used as the control. Treatments (mean ± SE, *n* = 10) with different letters are significantly different at *P* < 0.01, Tukey's HSD test. Comparative photomicrographs show live cells (prevent dye invasion) in PBS ingested *S. frugiperda* larvae and dead cells (stain dark blue because of disintegrated membrane) in Cry1AcF ingested *S. frugiperda* larvae. Scale bar = 10 μm.

PO. In view of this, hemolymph PO activity was measured in the test insects at 12 (in order to detect the early responses in host insect) and 24 h after Cry1AcF ingestion. Compared to PBS (0.05 OD$_{490}$/min/mg protein), a significantly ($F$ (2, 27) = 19.24, $P$ = 0.004) higher amount of PO induction (0.42–0.68 OD$_{490}$/min/mg protein) was documented in *S. frugiperda* larvae ingested with LD$_{50}$ and LD$_{90}$ doses of the toxin, at 12 h after inoculation (Fig. 4A). At 24 h, PO activity further increased ($F$ (2, 27) = 17.56, $P$ = 0.003) to 0.62–0.92 OD$_{490}$/min/mg protein in *S. frugiperda* (Fig. 4B). An identical trend was observed in *S. litura* larvae (Fig. 4).

## Cry1AcF intoxication altered the expression of gut receptors in *S. frugiperda* and *S. litura*

Initially, stage-specific and tissue-specific expression profiles of different Cry receptor-encoding genes was analyzed in *S. frugiperda* and *S. litura via* RT-qPCR. Transcripts of CAD, ABCC2, ALP1 and APN receptor were most abundantly expressed (*P* < 0.01) in the fourth- and fifth-instar stage compared to other life stages of *S. frugiperda* (Fig. S1). All the receptor gene mRNAs were greatly upregulated (*P* < 0.01) in the midgut tissues than other body parts including foregut, hindgut, Malpighian tubules, head and fat body of *S. frugiperda* fourth-instar stage (Fig. S2). Likewise, CAD, ABCC2, ALP1 and APN

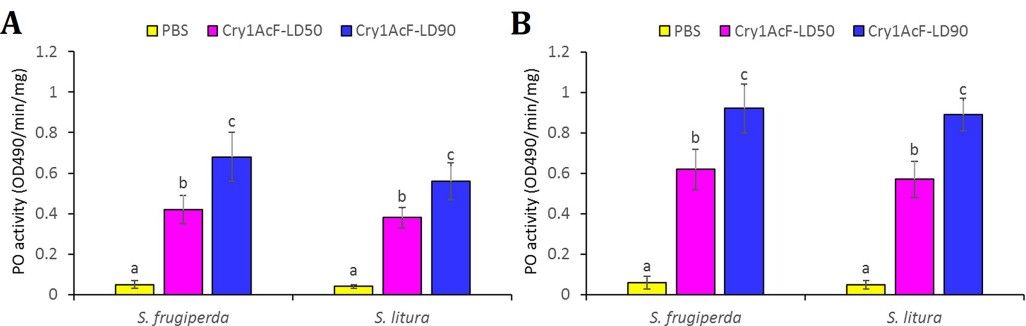

**Figure 4 Hemolymph phenoloxidase (PO) enzyme activity in *S. frugiperda* and *S. litura* fourth-instar larvae upon ingestion of Cry1AcF in LD$_{50}$ and LD$_{90}$ doses.** PO activity was measured at 12 (A) and 24 (B) h after toxin administration. PBS was used as the control. PO activity was measured using L-DOPA as substrate and change in absorbance (due to melanin synthesis) after 30 min was recorded at 490 nm. PO activity was expressed as OD$_{490}$ per minute per mg protein. PBS + L-DOPA was used as blank and its absorbance reading (at 490 nm) was subtracted from each sample. Treatments (mean ± SE, $n = 10$) with different letters are significantly different at $P < 0.01$, Tukey's HSD test.

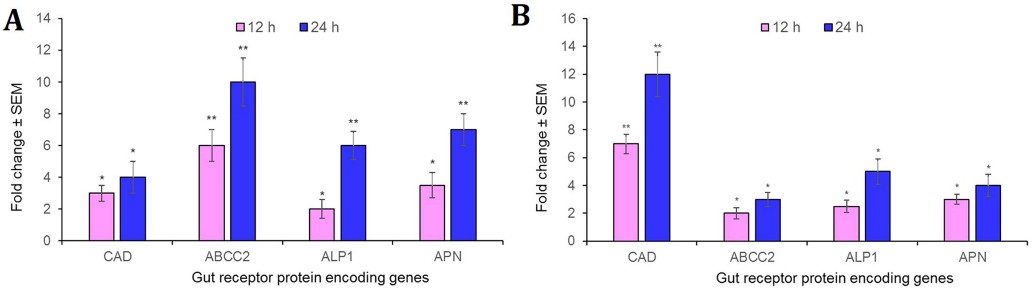

**Figure 5 Relative expression of receptor protein encoding genes in the midgut of *S. frugiperda*.** (A) and *S. litura* (B) fourth-instar larvae at 12 and 24 h after Cry1AcF ingestion (in LD$_{50}$ doses). The asterisk ($^*P < 0.01$, $^{**}P < 0.001$; Tukey's HSD test) is indicative of significant differential expression of target gene mRNAs compared to their baseline expression (fold change values set at 1) in insects ingested with PBS. Gene expression was normalized using endogenous reference genes of *S. frugiperda* (*β-actin* and *GAPDH*) and *S. litura* (*rps3*). Each bar represents the mean fold change value ± SE of RT-qPCR runs in five biological and three technical replicates.

transcripts were abundantly expressed ($P < 0.01$) in both fourth- and fifth-instar stages of *S. litura* (Fig. S3). All these receptor mRNAs were highly upregulated in the midgut tissues compared to other body parts of *S. litura* (Fig. S4).

Intriguingly, CAD, ABCC2, ALP1 and APN receptor-encoding genes were significantly ($P < 0.01$) overexpressed in the midgut tissues in *S. frugiperda* fourth-instar larvae at 12 h after oral administration of Cry1AcF toxin in LD$_{50}$ dose, compared to the expression levels of identical genes in PBS ingested larvae. At 24 h after toxin ingestion, expression level of these genes were further upregulated by several folds ($P < 0.001$) compared to their expression level during 12 h time period (Fig. 5A). A similar trend was observed with *S. litura* fourth-instar larvae. Most importantly, CAD transcripts were greatly expressed ($P < 0.001$) in *S. litura* at both 12 and 24 h after Cry1AcF ingestion (Fig. 5B). Herein, we assume that one or few of these receptor proteins may play crucial role during Cry1AcF intoxication process in *S. frugiperda* and *S. litura*. However, the possibility that the

upregulation of these receptor genes was an indirect response to the action of the Cry toxin in the midgut tissue cannot be dismissed because of our inability of using a non-receptor midgut protein as a control.

## RNAi of gut receptor-encoding genes altered *S. frugiperda* and *S. litura* susceptibility to Cry1AcF

Target dsRNA molecules (ideally 400–500 bp long sequences are readily processed by RNaseIII or Dicer enzyme) from the coding sequences of *S. frugiperda* (CAD: 411 bp, ABCC2: 406 bp, ALP1: 440 bp, APN: 410 bp) and *S. litura* (CAD: 427 bp, ABCC2: 447 bp, ALP1: 419 bp, APN: 442 bp) were designed according to the prediction of greater siRNA generation probabilities in the targeted region compared to untargeted stretches in the same sequence. Next, possible RNAi off-target effect was analyzed by using our target dsRNA sequences as query against *Drosophila melanogaster* siRNA database in DsCheck server (http://dscheck.rnai.jp/). Results indicated minimal homology of *D. melanogaster* siRNAs with that of siRNAs generated from dsRNAs of CAD, ABCC2, ALP1 and APN (data not shown). DsRNAs (cloned in L4440 vector) were expressed *via E. coli* HT115 (RNase III deficient) cells since ingestion of this recombinant bacteria can produce intact dsRNA molecules in the insect gut.

CAD, ABCC2, ALP1 and APN expression in the midgut tissues of *S. frugiperda* were considerably ($P < 0.01$) reduced by 75%, 61%, 79% and 65% in CAD, ABCC2, ALP1 and APN dsRNA-ingested larvae, respectively, compared to their expression in PBS-ingested larvae at 24 h after dsRNA treatment (Fig. S5; fold change in expression of a target gene in dsRNA-treated larvae was subtracted from that in control larvae and converted into percentage values). Similarly, CAD, ABCC2, ALP1 and APN expression in *S. litura* gut were markedly ($P < 0.01$) downregulated by 76%, 58%, 68% and 72% in CAD, ABCC2, ALP1 and APN dsRNA-ingested larvae, respectively, compared to control (Fig. S6). Ingestion of GFP dsRNA did not change ($P > 0.01$) the expression of receptor genes mRNAs in either of the insect species. In addition, transcript abundance of CAD was unchanged ($P > 0.01$) in ABCC2, ALP1 and APN dsRNA-treated insects. *Vice versa* was observed with other receptor mRNAs as well in either of the insect species (Figs. S5 and S6). These results has indicated the target-specific induced silencing of receptor-encoding genes in the midgut tissues. However, *S. frugiperda* CAD, ABCC2, ALP1 and APN shared 98%, 99%, 99% and 80% identity in the dsRNA-binding regions with homologous transcripts XM_035584870, OL955484, XM_035587634 and XM_022979826, respectively (Fig. 6A). *S. litura* CAD, ALP1 and APN shared 99%, 100% and 99% identity in the dsRNA-binding regions with homologous transcripts XM_022970523, XM_022967284 and XM_022969246, respectively (Fig. 6B). This suggests that RNAi of receptor-encoding genes might have silenced their allelic variants as well. Nevertheless, this could not be proved experimentally because of the limitations associated with designing specific primers, which would distinguish the two highly homologous gene transcripts.

RNAi-induced downregulation of CAD, ABCC2, ALP1 and APN transcripts in the midgut of *S. frugiperda* or *S. litura* did not confer any off-target effect on larval

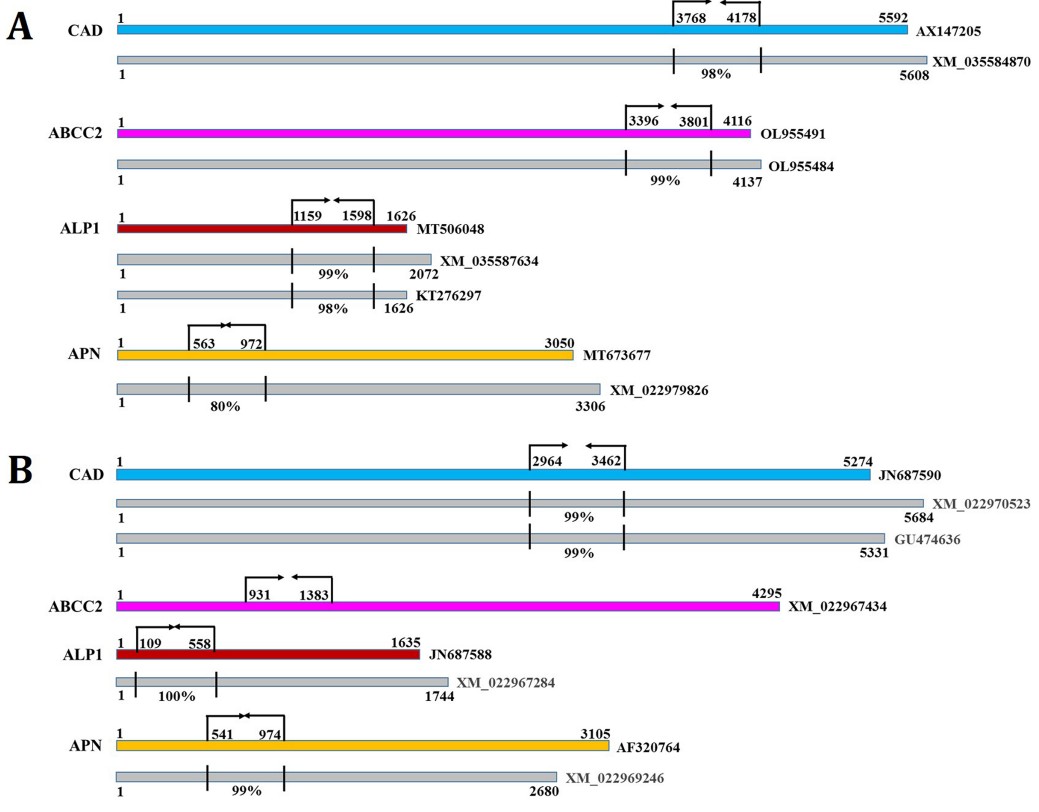

**Figure 6 The dsRNA binding sites (indicated by arrows and perpendicular lines) in the coding sequences of *S. frugiperda* (A) and *S. litura* (B) CAD, ABCC2, ALP1 and APN are shown.** Numbers indicate the sequence coordinates. The homologous transcripts (indicated by grey boxes) that were aligned with the target receptors are shown. Perpendicular lines indicate percent identity in the dsRNA target region. NCBI Genebank identifiers for different transcripts are listed at the end.

development and metamorphosis process as evidenced by the percent pupation and adult emergence data obtained at 7–10 days after dsRNA ingestion (Fig. S7).

Post 24 h of dsRNA ingestion, larvae were force-fed with Cry1AcF toxin in $LD_{50}$ and $LD_{90}$ doses followed by incubated in artificial diet. After another 24 h, larval mortality data was recorded. Interestingly, in CAD and ABCC2 silenced *S. frugiperda*, Cry1AcF (in both sub-lethal and lethal doses)-induced toxicity did not differ significantly ($P > 0.01$) with that of PBS- or GFP dsRNA-treated larvae (Fig. 7A). On the contrary, compared to control, Cry1AcF (in both $LD_{50}$ and $LD_{90}$ doses)-induced mortality in ALP1 and APN dsRNA pretreated *S. frugiperda* were significantly ($P < 0.01$) reduced by 42.85–43.39% and 46.15–49.06%, respectively (Fig. 7A). However, in ALP1 and APN silenced *S. litura*, Cry1AcF (in $LD_{50}$ and $LD_{90}$ doses)-induced toxicity did not differ significantly ($P > 0.01$) with that of PBS- or GFP dsRNA-treated larvae (Fig. 7B). Compared to control, a marginal yet significant ($P < 0.01$) reduction (28.57% and 21.05% in $LD_{50}$ and $LD_{90}$ doses, respectively) in *S. litura* mortality was documented when larvae were pretreated with ABCC2 dsRNAs (Fig. 7B). When CAD transcripts were knocked down in *S. litura*, a greatest ($P < 0.01$) reduction (55.36% and 52.63% in $LD_{50}$ and $LD_{90}$ doses, respectively) in

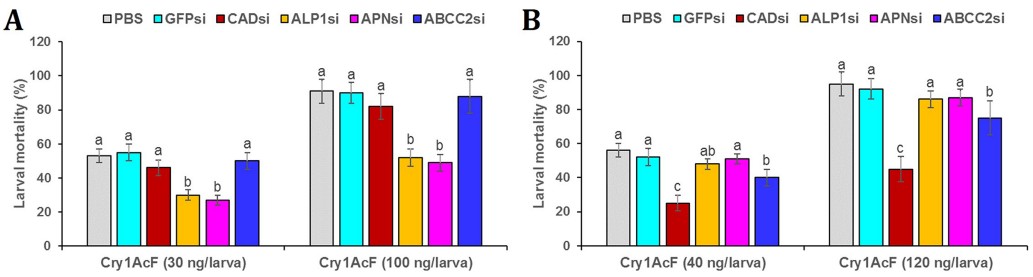

**Figure 7 RNAi-mediated silencing (si) of Cry receptor encoding genes differentially altered the susceptibility of *S. frugiperda* (A) and *S. litura* (B) to Cry1AcF toxin.** Initially, fourth-instar larvae were orally ingested with target receptor dsRNA-expressing *E. coli* HT115 cells. At 24 h of dsRNA ingestion, LD$_{50}$ (30 and 40 ng per larva for *S. frugiperda* and *S. litura*, respectively) and LD$_{90}$ (100 and 120 ng per larva for *S. frugiperda* and *S. litura*, respectively) doses of Cry1AcF were orally administered. After another 24 h percent mortality data was recorded. Different letters indicate treatments (mean ± SE) are significantly different at $P < 0.01$, Tukey's HSD test, $n = 50$. GFP dsRNA and PBS were used as the non-native and negative control, respectively.

larval mortality was documented in comparison to control larvae (Fig. 7B). Taken together, our data putatively indicates ALP1/APN and CAD as potential receptors for Cry1AcF intoxication in *S. frugiperda* and *S. litura*, respectively, because induced suppression of these receptor-encoding genes disrupted the larval susceptibility to Cry1AcF at both LD$_{50}$ and LD$_{90}$ doses.

## DISCUSSION

Chimeric toxins provide an important armory towards the diversification of Cry protein repertoire especially when insect pests are constantly developing resistance against narrow-spectrum Cry bio-pesticides or Bt crops globally (*Zafar et al., 2020*; *Chen et al., 2021*). A chimeric toxin Cry1AcF (generated by swapping domains of Cry1Ac and Cry1F; *Rathinam et al., 2017, 2019*) was transgenically expressed in different crop plants and exhibited toxicity against lepidopteran pests (*Keshamma et al., 2012*; *Ramu et al., 2012*; *Muralimohan et al., 2020*; *Ramkumar et al., 2020*). The synergistic effect of Cry1Ac and Cry1F in 1:1 ratio augmented the insecticidal efficacy of chimeric toxin by 26 times (*Ramu et al., 2012*). In the current study, a detailed *in vivo* pathogenesis assay of Cry1AcF was carried out in two notorious noctuid pests *S. frugiperda* (an invasive pest showing increased incidence under global climate change) and *S. litura*. Oral ingestion of Cry1AcF caused greater toxicity in the fourth-instar larvae of *S. frugiperda* (LD$_{50}$: 30 ng/larva; LD$_{90}$: 100 ng/larva) than *S. litura* (LD$_{50}$: 40 ng/larva; LD$_{90}$: 120 ng/larva) along with phenotypic alterations such as dead-like posture with extended prolegs and hemolymph melanization at 24 h post inoculation.

Histopathology of the midgut transverse sections showed Cry1AcF-induced extensive gut damage in both *S. frugiperda* and *S. litura* fourth-instar larvae. Compared to control larvae, toxin-ingested larvae exhibited disintegrated and sloughed midgut epithelial tissue into the midgut lumen followed by rupture of basement membrane and visceral muscle layer (at midgut-homecoel interface) putatively leading to leaky gut. An analogous histopathology data was previously documented in our laboratory when Cry1AcF toxin

was orally delivered into the greater wax moth *G. mellonella* larvae (*Dutta et al., 2022a*). Similarly, an orally active bacterial toxin TcaB (characterized from the entomopathogenic bacterium *Photorhabdus akhurstii*) caused lysis of the midgut epithelial cells followed by shedding of epithelium into the lumen leading to compromised gut homeostasis in *S. frugiperda*, *S. litura* and *H. armigera* fourth-instar larvae (*Dutta et al., 2021*). Because δ-endotoxin and vegetative insecticidal protein (Vip3A) of *B. thuringiensis* exhibit similar gut-active nature (*Adang, Crickmore & Jurat-Fuentes, 2014*; *Chakrabarty et al., 2022*), the pathogenesis process of Cry1AcF can be correlated with those toxins.

Upon reaching the body cavity *via* leaky gut, Cry1AcF might have targeted the hemocyte or immunocyte cells in order to cause toxicity. Toxin-like biomolecules frequently target the hemocyte cells and trigger apoptosis and cytolysis in order to modulate innate immune responses of host insects (*Whitten & Coates, 2017*). In the current study, at 24 h after oral ingestion, Cry1AcF elicited a sharp decline in total circulatory hemocyte numbers as well as hemocyte viability in *S. frugiperda* and *S. litura* fourth-instar larvae, compared to control larvae. Likewise, a substantial reduction in hemocyte counts because of Cry toxin-induced gut permeability was demonstrated in different insects (*Ericsson et al., 2009*; *Abd El-Aziz & Awad, 2010*; *Broderick, Raffa & Handelsman, 2010*; *Manachini et al., 2011*; *Grizanova et al., 2014*).

When challenge inoculated with toxin-like biomolecules, insects exhibit humoral immune response such as hemolymph melanization. This is resulted due to increased phenoloxidase enzyme activity in the hemolymph because hemocyte-bound pro-phenoloxidases are converted into phenoloxidases (*Whitten & Coates, 2017*). In the present study, Cry1AcF ingestion markedly elevated the hemolymph phenoloxidase activity in *S. frugiperda* and *S. litura* fourth-instar larvae, compared to control larvae. Intriguingly, phenoloxidase activity was initiated during early pathogenesis (12 h after Cry1AcF inoculation) and was further increased to manifold during late pathogenesis (24 h). Our results suggest that, Cry1AcF can be used as an immunomodulatory or immune-stimulatory agent in insect defense response studies. Notably, a 3-day sustaining Cry-induced increased hemolymph phenoloxidase activity was documented in *G. mellonella* (*Grizanova et al., 2014*). Similar data was documented in *H. armigera* and *Trichoplusia ni* upon challenge inoculation with different Cry toxins (*Rahman et al., 2004*; *Ma et al., 2005*; *Ayres & Schneider, 2008*). *Cerenius, Lee & Soderhall (2008)* speculated that elevated phenoloxidase activity is resulted due to induced gut repair process, which limits the toxin escape into body cavity through Cry-elicited perforated gut. Nevertheless, other studies argued that Cry-elicited immunomodulation prevents host insect resistance to further Cry intoxication in the intestine (*Kwon & Kim, 2007*; *Broderick, Raffa & Handelsman, 2010*; *Richards & Dani, 2010*). The initiation of gut repair process was not observed in either of the test insects ingested with Cry1AcF in our histopathology experiments.

CAD, ABCC (transmembrane proteins), APN and ALP (tether to the epithelial cell membrane *via* GPI-anchors) are known receptor proteins in the brush border cells that are crucial for Cry binding and intoxication mechanism in the intestine (*Bravo et al., 2011*; *Fabrick & Wu, 2015*; *Liu et al., 2021*). This prompted us to analyze whether Cry1AcF

intoxication in *S. frugiperda* and *S. litura* is dependent on binding to these receptors or not by deploying RNAi-mediated knockdown of individual receptor-encoding genes. Initially, developmental stage-and tissue-specific transcription patterns of CAD, ABCC2, ALP1 and APN in test insects was analyzed by RT-qPCR. All the receptor gene transcripts were greatly expressed in both fourth- and fifth-instar stages of *S. frugiperda* and *S. litura*. Additionally, all the receptor mRNAs were most abundantly expressed in the midgut tissues of *S. frugiperda* and *S. litura* fourth-instar larvae. In our earlier study, *S. frugiperda* and *S. litura* fourth-instar larval stages were found to be highly vulnerable to a bacterial toxin TcaB (*Dutta et al., 2021*).

Oral ingestion of bacterially-expressed dsRNAs of CAD, ABCC2, ALP1 and APN downregulated (compared to control) the transcription of corresponding receptor mRNAs by 75–76%, 58–61%, 68–79% and 65–72%, respectively, in the midgut tissue of *S. frugiperda* and *S. litura* fourth-instar larvae. Probability of any cross knockdown effect among the receptor mRNAs was non-existent as the sequence similarly between targeted dsRNA sequences of CAD, ABCC2, ALP1 and APN within the same test insect species was negligible (data not shown). In addition, silencing of receptor genes did not cause any negative effect on the growth and metamorphosis of either of the test insects.

Target-specific induced downregulation of receptor mRNAs variably altered the insect susceptibility to Cry1AcF toxin in our study. For example, the susceptibility of ALP1 and APN dsRNA-treated *S. frugiperda* larvae was decreased (compared to control) by 42.85–43.39% and 46.15–49.06%, respectively, when ingested with Cry1AcF toxin in $LD_{50}$ and $LD_{90}$ doses. On the contrary, susceptibility of CAD and ABCC2 dsRNA-treated *S. frugiperda* was alike of control larvae when ingested with Cry1AcF in lethal and sub-lethal doses. In ALP1 and APN dsRNA-treated *S. litura*, larval susceptibility was alike of control when ingested with Cry1AcF in $LD_{50}$ and $LD_{90}$ doses. Conversely, susceptibility of CAD dsRNA-treated *S. litura* was decreased (compared to control) by 55.36% and 52.63%, upon ingestion with Cry1AcF in $LD_{50}$ and $LD_{90}$ doses, respectively. Additionally, *S. litura* larvae pre-treated with ABCC2 dsRNA showed a meagre yet significant reduction (21.05–28.57%) in insect vulnerability to Cry1AcF (in $LD_{50}$ and $LD_{90}$ doses), compared to control. Taken together, our results indicate ALP1/APN and CAD/ABCC2 as the functional receptor for Cry1AcF toxicity in *S. frugiperda* and *S. litura*, respectively. Notably, either of the ALP, APN and ABCC receptors are involved in pore formation model of Cry mechanism of action. In this model, Domain I of Cry protein attaches into the brush border cell membrane *via* its hydrophobic helical hairpin, Domain II identifies receptor molecules and Domain III aids in receptor binding and pore formation (*Pacheco et al., 2009*; *Pardo-López, Sóberon & Bravo, 2013*; *Adang, Crickmore & Jurat-Fuentes, 2014*; *Jin et al., 2021*; *Jurat-Fuentes, Heckel & Ferré, 2021*). In the signaling pathway model, Cry binding to CAD receptor triggers adenylate cyclase activation that stimulates increased cAMP and protein kinase synthesis. This leads to an ion channel (formed in the brush border cell membrane)-dependent cellular signaling cascading effect, which ultimately causes cell death (*Zhang et al., 2006*; *Castella et al., 2019*).

## CONCLUSIONS

In our study, the chimeric toxin Cry1AcF conferred potent insecticidal activity in *S. frugiperda* and *S. litura*. Hemolymph melanization in the test insects was the resultant effect of Cry1AcF-induced gut damage followed by cytotoxicity in the hemocyte cells. Further, RNAi-mediated silencing of gut receptor encoding genes differentially altered the susceptibility of test insects to Cry1AcF toxin, indicating the potential role of these receptor proteins in Cry1AcF intoxication process. More specifically, our results indicated ALP1/APN and CAD/ABCC2 as the functional receptor for Cry1AcF toxicity in *S. frugiperda* and *S. litura*, respectively.

## ACKNOWLEDGEMENTS

We acknowledge Dr. S. Subramanian and Dr. D. Sagar, Division of Entomology, ICAR-Indian Agricultural Research Institute for providing us the laboratory space for rearing insects and conducting RNAi experiments.

### Funding

This work was supported by the Science and Engineering Research Board (SERB)-Department of Science and Technology (DST), Government of India (Project code: YSS/2014/000452). The funders had no role in study design, data collection and analysis, decision to publish, or preparation of the manuscript.

### Grant Disclosures

The following grant information was disclosed by the authors:
Science and Engineering Research Board (SERB).
Department of Science and Technology (DST).
Government of India: YSS/2014/000452.

### Competing Interests

Tushar K. Dutta is an Academic Editor for PeerJ.

### Author Contributions

- Tushar K. Dutta conceived and designed the experiments, performed the experiments, analyzed the data, prepared figures and/or tables, authored or reviewed drafts of the article, and approved the final draft.
- Kodhandaraman Santhoshkumar performed the experiments, prepared figures and/or tables, and approved the final draft.
- Arudhimath Veeresh performed the experiments, prepared figures and/or tables, and approved the final draft.
- Chandramani Waghmare performed the experiments, authored or reviewed drafts of the article, and approved the final draft.

- Chetna Mathur performed the experiments, prepared figures and/or tables, and approved the final draft.
- Rohini Sreevathsa performed the experiments, authored or reviewed drafts of the article, and approved the final draft.

## Data Availability

Raw data is available in the Supplemental Files.

## Supplemental Information

Supplemental information for this article can be found online at http://dx.doi.org/10.7717/peerj.14716#supplemental-information.

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
