# Peer review of "RNAi-based knockdown of candidate gut receptor genes altered the susceptibility of Spodoptera frugiperda and S. litura larvae to a chimeric toxin Cry1AcF"

_PeerJ, doi:10.7717/peerj.14716_

## Round 0.1 · original submission · Minor Revisions

Although the current decision is minor revisions, I would urge the authors to take all the comments from all reviewers as major comments that need to be addressed before the final decision is made. Reviewer 1 made some valid and strong points and please address in the revision

Reviewer 1 ·

Basic reporting

The manuscript shows the use of Cry toxins could be a control method alternative to reduce the survivorship of Spodoptera frugiperda and S. litura, both insect pests. Moreover, the authors analized the histopathology of the midggus from those insects to understand how the Cry toxins affect the cells in the midgut. And the author perfomed RNAi assays to understand if the receptor-encoding genes play important roles with Cry toxins. The manuscript shows relevant results and to hypotheses.
The manuscript is very well written and technically correct. I listed some changes below that authors should do. And the authors should include more literature references. The figures and tables are very well described.

- Line 30_ Remove “bacterium”
- line 32: write which means Bt before here
- Line 33: Change “avert” to avoid
- Lines 33-34: Change “protein-engineered chimeric toxin-based strategy was advocated” to “protein-engineered chimeric toxin-based strategy have been analyzed”
- Line 40: Rewrite (What do you mean?)
- Line 45: Change “…we performed RNAi (bacterially-expressed dsRNAs were used)-mediated silencing…” to “…we performed RNAi mediated by bacterially-expressed dsRNAs…”
- Line 76: Remove “bacterium”
- Line 77: You should add more literature references, not only Chakrabarty et al. 2022. (You repeat many times Chakrabarty et al. 2022…)
- Line 129: Change “inoculation” to “oral ingestion”
- Line 134: Change “inoculation” to “oral ingestion”
Line 135: Remove “another”
*Some moments the text is in “we” and other “passive voice”
- Line 149: Change “inoculation” to “oral ingestion”
- Line 149: Change “10 µL sample of fresh hemolymph” to “Sample of fresh hemolymph (10 µL)…”
- Line 192: Where is the Tables 1 and 2 in Supplementary material?
- Line 220: Remove “another”

Experimental design

The experimental design is very well desbribed and there are sufiicient details and intofmation to replicate.

Validity of the findings

The results found by the authors are very important to prove in which gut receptor-proteins binds in Cry toxins binds to the . Moreover, the work shows the Cry toxins-effects in the Spodoptera frugiperda and S. litura (survivorship, anatomy, immune response and gut histopathology). Thus, the Cry toxins could be used as a control method of both insect pest.

·

Basic reporting

The article “RNAi-based knockdown of candidate gut receptor genes altered the susceptibility of Spodoptera frugiperda and S. litura larvae to a chimeric toxin Cry1AcF” is aimed to the interesting and popular topic of insect resistance to Bt.
Clear and unambiguous, professional English used throughout.
Literature references, sufficient field background/context provided.
Professional article structure, figures, tables.

Experimental design

The work was done with some modern methods (RNAi), good number of insects for THC and PO activity, especially for gene expression (5). The MS is written well, results are well illustrated, the discussion is nice to read, references are suitable for the discussed questions.

Validity of the findings

All underlying data have been provided; they are robust, statistically sound, & controlled.
Conclusions are well stated, linked to original research question & limited to supporting results

Additional comments

Some suggestions authors could find below.
1. M&M. Please add the reference to the methods of Histopathology study (line 133), Total hemocyte counts (line 148-152), hemocyte viability (line 152-253); PO (line 164).
2. M&M. Please, add full name of PBS and information about PBS components (Line 151).
3. M&M. Please, add full information about Anticoagulant components (Line 151).
4. M&M. Please, add full name of l-DOPA.
5. M&M. Please, add protocol and reference about method of protein measurement in midgut samples (Line 164).
6. M&M. Please, add information about calculation of RNAi-induced downregulation (Line 223).
7. M&M. to be clearer for the readers, please, add the method for statistical analysis of THC and PO. Was the distribution of PO data being normal?
8. Please, explain how the reduction in expression was calculated in percentage (Line 328-334).

·

Basic reporting

The manuscript entitled “RNAi-based knockdown of candidate gut receptor genes altered the susceptibility of Spodoptera frugiperda and S. litura larvae to a chimeric toxin Cry1AcF” presented to PeerJ describes a characterization of the toxic effects of Cry1AcF (chimeric toxin) in S. frugiperda and S. litura and attempts to identify toxin receptor candidates by RNAi-mediated gene knock down. Most experiments appear to be well conducted. The authors repeat the same methodology as in previous works (Dutta et al., 2022a; 2022b) with the model insect, Galleria mellonella, now in important pests. However, the manuscript cannot be accepted in its present form. I have raised a few questions that must be carefully addressed before reconsidering.

Experimental design

Introduction
- More information about both Spodoptera species is needed in the introduction (importance, which crops they attack, losses, etc);
- Authors described several putative Cry toxin receptors for different species, however they did not mention any for both studied species. Please, add this information in the introduction.


Some points of experimental design should be clarified or corrected:

- Line 115, the word “sick” is not appropriate for crops injured by insects. Use the word “infested” instead.
- Lines 117-120, there is a confusion of information about the insect rearing. First is said that neonate larvae were reared in castor bean leaves, after is said that Adults and larvae were reared with artificial diet. Which one is correct? The artificial diet for adults and larvae should be different, since they have different feeding habits.
- Line 127, Why the authors did not use a known Cry toxin that that is effective against both pests? I cannot see how we can compare both toxins mortality.
- Lines 134 to 138, Were the larva dead or alive? If they were alive, why not dissect the fresh midgut and then fix the tissue?
- Line 184, Why only one reference gene was used for S. frugiperda gene expression normalization? According to the MIQE guidelines (https://doi.org/10.1373/clinchem.2008.112797) at least two reference genes should be used to properly analyze qPCR data.
- Lines 209-211, Please describe how dsRNA was extracted from E. coli cells.

Validity of the findings

The findings are not that different from previous works, but I believe they are still valid, as they were validated in important agricultural pests.

Points to be clarified:
- LD50 and 90 for Cry1AcF are not different from those of TcaB since the values of Confidence Interval are overlapping (Table 1). LD values without confidence interval are not comparable, then the values of LD in ng/g of insect are not valid. I suggest to authors adjusting the text and remove the values of LD50 and 90 expressed in ng/g of insect.
- Lines 294-314, I wonder why all the genes tested were upregulated when the insects were fed with Cry1AcF. I believe that the authors need, as a control, to test other genes expressed in the midgut that are not related to Cry toxin activity, to see if this upregulation is a direct or indirect response to Cry toxin action.
- Lines 436-437, The sentence seems to be displaced.

Additional comments

No additional comments

---

## Round 0.2 · Minor Revisions

Thank you for revising the submission. Please address the remaining concerns of Reviewer 3.

·

Basic reporting

The authors have made several improvements to the manuscript, and I am satisfied with most of them. However, I am still not convinced by the authors' responses to two of the issues raised.

Initially, in the results described in Table 1, the authors still state that the LD50 values of Cry1AcF and TcaB are different from each other, even when the confidence intervals overlap. To calculate the LD50 in ng/g of larvae, the authors must perform the Probit analysis again. LD50 or 90 values without confidence interval are not comparable. Apparently, Guo et al. (2018, Toxicon 150: 297-303) used a different methodology to calculate the LD50. So, it could not be used to justify the calculations presented in this manuscript.

Finally, about the comment:

'- Lines 294-314, (lines 305-325 on version R1) I wonder why all tested genes were up-regulated when insects were fed Cry1AcF. I believe that the authors need, as a control, to test other genes expressed in the midgut that are not related to the activity of the Cry toxin, to see if this upregulation is a direct or indirect response to the action of the Cry toxin.'

I understand that the authors could not analyze another midgut gene as a control this time. However, it should be mentioned, in the discussion, the possibility that the upregulation of these genes is an indirect response to the action of the Cry toxin in the midgut.

Experimental design

no comment

Validity of the findings

no comment

Additional comments

no comment

---

## Round 0.3 · accepted · Accept

Hello,

Your submission has been accepted for publication with PeerJ after careful review. Congratulations!!

·

Basic reporting

All the raised points were addressed properly.

Experimental design

No comment

Validity of the findings

No comment

Additional comments

No comment